# Predictors of Seasonal Influenza Vaccination Willingness among High-Risk Populations Three Years after the Onset of the COVID-19 Pandemic

**DOI:** 10.3390/vaccines11020331

**Published:** 2023-02-01

**Authors:** Aglaia Katsiroumpa, Panayota Sourtzi, Daphne Kaitelidou, Olga Siskou, Olympia Konstantakopoulou, Petros Galanis

**Affiliations:** 1Clinical Epidemiology Laboratory, Faculty of Nursing, National and Kapodistrian University of Athens, 11527 Athens, Greece; 2Faculty of Nursing, National and Kapodistrian University of Athens, 11527 Athens, Greece; 3Center for Health Services Management and Evaluation, Faculty of Nursing, National and Kapodistrian University of Athens, 11527 Athens, Greece; 4Department of Tourism Studies, University of Piraeus, 18534 Piraeus, Greece

**Keywords:** influenza, vaccination, willingness, COVID-19, intention

## Abstract

High-risk populations are at increased risk of severe influenza-related illness, hospitalization, and death due to influenza. The aim of our study was to assess the willingness of high-risk populations to take the influenza vaccine for the 2022–2023 season, and to investigate the factors associated with such willingness. We conducted a cross-sectional study in Greece in September 2022 using a convenience sample. We considered demographic characteristics, COVID-19-related variables, resilience, social support, anxiety, depression, and COVID-19-related burnout as potential predictors. Among participants, 39.4% were willing to accept the seasonal influenza vaccine, 33.9% were unwilling, and 26.8% were hesitant. Multivariable analysis identified that increased age and increased family support were associated with increased influenza vaccination willingness. Moreover, participants that have received COVID-19 booster doses were more willing to accept the influenza vaccine. In contrast, adverse effects because of COVID-19 vaccination and exhaustion due to measures against COVID-19 reduced influenza vaccination willingness. We found that the intention of high-risk populations to receive the influenza vaccine was low. Our study contributes to an increased understanding of the factors that affect vaccination willingness. Public health authorities could use this information to update vaccination programs against influenza. Emphasis should be given on safety and effectiveness issues.

## 1. Introduction

Influenza has a significant impact on elderly and chronic disease patients [1]. Additionally, influenza requires medical treatment more often and causes more serious complications in individuals with chronic medical conditions (e.g., cardiovascular, pulmonary, metabolic, hematologic, or renal diseases). For instance, influenza complications are three times more among chronic disease patients compared to healthy people [2]. Moreover, about 389,000 respiratory deaths are associated with influenza worldwide each year and 67% of these are among people over 65 years old [3]. Moreover, hospitalization rates are the highest in adults older than 65 years [4].

Effects of influenza vaccination are especially considerable for chronic disease patients by reducing related hospitalizations and all-cause mortality. For example, a meta-analysis showed that influenza vaccination led to a significant 28% reduction in major adverse cardiovascular events and a 41% reduction in cardiovascular disease mortality among patients with cardiovascular diseases [5]. Additionally, according to a recent meta-analysis, influenza vaccination could act as a protective factor by reducing the risk of dementia among older people by 29% [6]. Moreover, two systematic reviews showed that influenza vaccination might be cost-effective and cost-saving for elderly and people with chronic medical conditions (such as chronic obstructive pulmonary disease patients) [7,8].

The literature suggests that influenza vaccination might be beneficial to SARS-CoV-2 infection and to the clinical outcomes among COVID-19 patients. In particular, meta-analyses showed that influenza vaccination reduced the risk of SARS-CoV-2 infection up to 24%, hospitalization up to 29%, intensive care unit admission up to 37%, and death up to 32% [9,10,11,12]. However, we should notice that the evidence is not sufficient to affirm the positive effect of influenza vaccination. Prospective, bigger, and more valid studies should be conducted in order to get more solid results.

During the COVID-19 pandemic, only four studies [13,14,15,16] assessed the intentions of high-risk populations to receive the influenza vaccine for the 2020–2021 season. Additionally, these studies investigated only demographic variables and previous influenza vaccine acceptance as potential predictors. Moreover, several other studies [17,18,19,20,21] determined the influenza vaccination intention for the 2020–2021 and 2021–2022 seasons but samples were recruited from the general population or healthcare workers.

Additionally, there are significant differences between the European countries regarding the influenza vaccination coverage rate among the elderly. In particular, data from “The Organization for Economic Co-operation and Development (OECD)” show that the influenza vaccination coverage rate for people over 65 years for the 2020–2021 season was 22.3% in Hungary, 62.7% in Finland, 65.3% in Greece and Italy, 69% in Spain, 75.4% in Ireland, and 78% in Denmark [22]. Moreover, a recent study in European countries found that the influenza vaccination coverage rate for elderly for the 2020–2021 season was 59.9% in France, 65.3% in Italy, 67.7% in Spain, 67.9% in the Netherlands, and 80.9% in England [23]. In Greece, two studies including healthcare workers found influenza vaccination coverage rate as 74% and 76% for the flu season 2020–2021 [24,25].

Understanding the factors that promote the intention of individuals to get vaccinated is crucial to plan and tailor future immunization programs for the influenza virus. Even more important is the fact that certain predictors of vaccination willingness could be modified in order to increase the influenza vaccination coverage rate. For example, knowledge and beliefs about vaccine safety and effectiveness, perceptions of vaccination as an effective coping strategy, influence of social pressure, and information about vaccination from official public health authorities are modifiable factors which are associated with higher willingness and uptake of vaccination [17,26].

Thus, the aim of our study is to estimate the willingness of high-risk populations in Greece to take the influenza vaccine for the 2022–2023 season, and to identify factors that influence such willingness.

## 2. Materials and Methods

### 2.1. Study Design and Population

Greek Ministry of Health provided free influenza vaccine to all citizens from 03 October 2022. We conducted a cross-sectional study in Greece in September 2022 in order to assess the intention of high-risk groups to take the influenza vaccine just before the free distribution of the vaccine. 

Our study population included individuals that are at higher risk of developing severe disease according to the European Centre for Disease Prevention and Control [27] and the Centers for Disease Control and Prevention [28]. Thus, our inclusion criteria were individuals being over 65 years old or adults with certain chronic medical conditions (asthma, diabetes mellitus, heart disease, stroke, chronic kidney disease, chronic obstructive pulmonary disease, liver disorders, neurologic and neurodevelopment conditions, patients with a weakened immune system due to hematological conditions and HIV infection). Moreover, we included only individuals that had previously been fully vaccinated against COVID-19 with the primary doses (two doses with Pfizer/Moderna vaccines or one dose with Johnson & Johnson vaccine). In addition, the study questionnaire was in Greek. Thus, people who understand Greek language participated in our study. We excluded people younger than 18 years and pregnant women, as pregnancy is a risk factor for potentially serious flu complications. We considered that pregnant women are a population with large differences in relation to the elderly and patients with a chronic health condition. Therefore, we decided not to include pregnant women in our study.

We created an online form of the study questionnaire using Google forms. Then, we disseminated the questionnaire through social media and our email contacts. Thus, we obtained a convenience sample with an unknown response rate. We collected data in an anonymous way since we did not obtain participants’ personal data, e.g., their name. The first page of the questionnaire informed individuals about the aim and the design of our study. Then, we asked people if they are over 65 years old or if they are suffering from a chronic disease (see above) in order to meet our inclusion criteria. After a positive answer, we asked respondents to give their informed consent in order to participate in the study. Afterwards, participants could complete the study questionnaire. Therefore, participation was anonymous and voluntary. 

Minimum sample size required 219 individuals considering a low effect size (f^2^ = 0.06), the precision level as 5% (alpha level), the power as 95%, and the number of predictors as 16 [29].

We applied the guidelines of the Declaration of Helsinki in order to perform our study. Moreover, the study protocol was approved by the Ethics Committee of Faculty of Nursing, National and Kapodistrian University of Athens (reference number; 370, 2 September 2021).

### 2.2. Measures

We measured demographic characteristics, COVID-19-related variables, resilience, social support, anxiety, depression, COVID-19-related burnout, and influenza vaccination willingness.

In particular, demographic characteristics included gender (males or females), age (continuous variable), educational level (elementary school, high school or university degree), and self-perceived health status (very poor, poor, moderate, good, very good). 

COVID-19-related variables included SARS-CoV-2 infection (no or yes), COVID-19 booster doses (no or yes), and adverse effects because of COVID-19 vaccination (scale from 0 [no adverse effects] to 10 [many adverse effects]).

We used the Brief Resilience Scale (BRS) to measure resilience among participants [30]. BRS comprises six items and overall score ranges from 1 (low resilience) to 5 (high resilience). BRS is a reliable and valid tool in Greek [31]. Cronbach’s alpha for the scale was 0.814 in our study.

We used the Multidimensional Scale of Perceived Social Support (MSPSS) to measure social support among participants [32]. MSPSS comprises 12 items and three factors: family support, friends support, and support from significant others. Each factor takes a value from 1 to 7 with higher values indicating higher level of support. Previous research proves the reliability and the validity of the MSPSS in Greek [33]. In our study, Cronbach’s alpha for family support, friends support, and support from significant others was 0.931, 0.939, and 0.898 respectively.

We used the Patient Health Questionnaire-4 (PHQ-4) to measure anxiety and depression among participants [34]. PHQ-4 comprises four items and two factors: anxiety and depression. Scores on anxiety and depression range from 0 (normal) to 6 (severe symptoms). PHQ-4 has been validated in Greek [35]. In our study, Cronbach’s alpha for anxiety was 0.893 and for depression was 0.826.

We measured COVID-19-related burnout that participants experience with the COVID-19 burnout scale (COVID-19-BS) [36]. COVID-19-BS consists of 13 items and comprises three factors: emotional exhaustion, physical exhaustion, and exhaustion due to measures against COVID-19. Score on each factor ranges from 1 (low level of burnout) to 5 (high level of burnout). Literature supports the reliability and validity of COVID-19-BS in Greek [37]. In our study, Cronbach’s alpha for family support, friends support, and support from significant others was 0.910, 0.868, and 0.889 respectively.

We used the question “How likely do you think you are to get the seasonal influenza vaccine?” to measure influenza vaccination willingness among the participants. Answers were on a scale from 0 (very unlikely) to 10 (very likely). Moreover, we used a priori cut-off points in order to categorize participants as willing (score from 8 to 10), hesitant (score from 3 to 7), and unwilling (score from 0 to 2).

### 2.3. Statistical Analysis

We use numbers and percentages to describe categorical variables. Moreover, we use mean, standard deviation, median, minimum value, and maximum value to describe continuous variables. We excluded from analysis respondents with no data on more than 20% of study variables. In that case, 12 respondents were excluded from the analysis. We used demographic characteristics, COVID-19-related variables, resilience, social support, anxiety, depression, and COVID-19-related burnout as the independent variables. Influenza vaccination willingness score was the dependent variable. We performed univariate and multivariable linear regression analysis to assess the impact of independent variables on the influenza vaccination willingness score. In the multivariable linear regression model, we included all the independent variables simultaneously in order to eliminate confounding. Moreover, we performed a sensitivity analysis removing strong predictors of vaccination willingness which we found in the main analysis. For regression models, we presented unadjusted and adjusted coefficients beta, 95% confidence intervals (CI), and *p*-values. Variables with a *p*-value less than 0.05 in the multivariable linear regression model were considered as statistically significant. We used the IBM SPSS 21.0 (IBM Corp. Released 2012. IBM SPSS Statistics for Windows, Version 21.0. Armonk, NY, USA: IBM Corp.) for the analysis.

## 3. Results

Most of the participants were females (78.7) with a university degree (78%). Mean age was 44.9 years. Among participants, 59.8% reported a good health status, 20.5% a very good, 18.1% a moderate, and 1.6% a poor. About COVID-19-related variables, 64.6% of participants had previously been infected by SARS-CoV-2 during the pandemic and 89.8% have received COVID-19 booster doses. Moreover, mean score of adverse effects because of COVID-19 vaccination was 2.9 indicating that the frequency of occurrence of the adverse effects was low. Demographic and COVID-19-related variables of the study population are presented in Table 1.

Detailed statistics for the scales used in the study are presented in Table 2. Among participants, 39.4% (*n* = 100) were willing to accept the seasonal influenza vaccine, 33.9% (*n* = 86) were unwilling, and 26.8% (*n* = 68) were hesitant. Moreover, mean willingness score was 5.1 indicating a moderate level of flu vaccine acceptance.

Resilience among participants was moderate with a mean score of 3.41. Moreover, anxiety and depression were moderate with mean scores of 2.41 and 2.26 respectively. Levels of support were high among participants. Moreover, support from significant others (mean score = 5.94) and family (mean score = 5.84) was higher than support from friends (mean score = 5.61). 

Participants experienced moderate levels of burnout due to the pandemic. Moreover, COVID-19-related emotional exhaustion (mean score = 3.41) and exhaustion due to measures against COVID-19 (mean score = 3.36) were higher than physical exhaustion (mean score = 2.53).

Results from linear regression analysis with influenza vaccination willingness score as the dependent variable are shown in Table 3. Multivariable analysis identified that increased age (adjusted coefficient beta = 0.040, 95% CI = 0.002 to 0.078) and increased family support (adjusted coefficient beta = 0.642, 95% CI = 0.181 to 1.103) were associated with increased influenza vaccination willingness. Moreover, participants that have received COVID-19 booster doses (adjusted coefficient beta = 1.841, 95% CI = 0.333 to 3.350) were more willing to accept the influenza vaccine. On the other hand, adverse effects because of COVID-19 vaccination (adjusted coefficient beta = −0.211, 95% CI = −0.407 to −0.016) and exhaustion due to measures against COVID-19 (adjusted coefficient beta = −0.561, 95% CI = −0.986 to −0.137) reduced influenza vaccination willingness.

Since the effect of COVID-19 booster doses was strong, we performed a sensitivity analysis by removing this variable from the final multivariable model (Appendix A). Sensitivity analysis confirmed the results from the first multivariable model (Table 3) with slight differences in the adjusted coefficients beta: age (adjusted coefficient beta = 0.045, 95% CI = 0.008 to 0.083); family support (adjusted coefficient beta = 0.687, 95% CI = 0.223 to 1.152); adverse effects because of COVID-19 vaccination (adjusted coefficient beta = −0.235, 95% CI = −0.432 to −0.038); exhaustion due to measures against COVID-19 (adjusted coefficient beta = −0.581, 95% CI = −1.009 to −0.153).

## 4. Discussion

To the best of our knowledge this is the first study that estimates the willingness of high-risk populations to be vaccinated against seasonal influenza for the 2022–2023 season and examines the factors which influence such willingness. Moreover, our study investigated for first time the impact of COVID-19-related variables (e.g., COVID-19-related burnout) on individuals’ willingness to accept the influenza vaccine.

Among our participants, 39.4% intended to be vaccinated against flu, 33.9% indicated they were unlikely to have the vaccination, and 26.8% were unsure. In our study, intention rate was low considering that vaccination uptake rate against influenza for people aged 65 and over was 56.2% in Greece 2018 [38]. Moreover, our intention rate is far below the WHO goal of 75% vaccination rate against seasonal influenza for elderly [38]. A similar influenza vaccination coverage for the older people is set as a goal by the 2009 European Union Council Recommendation [39]. Similar studies in Greece before the COVID-19 pandemic showed a vaccination uptake of 56.6% for people over the age of 60 [40] and of 34.8% [41] to 37.9% [42] for high-risk populations. Moreover, vaccination intention rate against influenza for the 2020–2021 season among high-risk people in Spain, Ireland, Japan, and USA was much higher compared to that in our study ranging from 43.3% to 79.7% [13,14,15,16]. Moreover, such intention rate was higher in studies including patients in United Kingdom (76.3%) [43] and China (54%) [44] before the pandemic. In addition, studies including samples from the general population presented a vaccination intention rate from 40.8% to 74.3% for the 2020–2021 and 2021–2022 seasons [17,18,19,20,21].

In our study, low levels of participants’ willingness to accept the influenza vaccination could be attributed to fatigue among people to adhere to preventive measures three years after the onset of the pandemic. A systematic review has already shown that adherence to preventive measures against COVID-19 is decreasing over time [45]. As COVID-19 pandemic continues to evolve, individuals’ fear of the disease is decreasing and that can result in less adherence to preventive measures such as vaccination, mask wearing, hand washing, etc. Early studies among nurses showed that higher levels of exhaustion due to measures against COVID-19 reduced influenza vaccination intention [46], while increased compliance with hygiene measures was associated with increased COVID-19 booster dose hesitancy [47]. Moreover, vaccine hesitancy could be another factor that contributes to low willingness in our participants to get vaccinated against influenza. Vaccine hesitancy is a well-known phenomenon, and World Health Organization recognizes it as one of the top ten threats to global health in 2019 [48]. Especially during the pandemic, lack of accurate knowledge about vaccines, confusing messages about vaccine safety and side effects, anti-vaccine myths, conspiracy beliefs, mistrust and suspicion of medical companies, and reduced trust in government and healthcare professionals increase vaccine hesitancy to COVID-19 and influenza vaccines [49,50,51,52].

Our findings showed that participants who experienced more adverse effects because of COVID-19 vaccination were less likely to accept the influenza vaccine. A systematic review of influenza vaccine hesitancy before the pandemic confirms this result since it found that adverse effects after previous influenza vaccination reduce intention of individuals to accept future vaccination [53]. In addition, a survey among European countries in 2019 found that 10% of people aged 65 and over did not take the influenza vaccine because they thought that vaccines could have adverse effects and are not safe [39]. Moreover, a systematic review including studies during the 2009 pandemic H1N1 influenza outbreak found that concerns about the safety of the vaccine and its side effects reduced vaccination uptake [26]. During the first year of the pandemic, studies in the USA [54], China [20], and Japan [14] found that concerns regarding adverse effects from the influenza vaccine and previous negative vaccination experiences discourage people to accept future influenza vaccination. In addition, a meta-analysis showed that people who experience more adverse effects and discomfort after previous COVID-19 vaccination were less likely to accept a booster dose [55].

According to our results, increased family support among participants was associated with increased vaccination willingness. The literature supports this finding since several studies before the pandemic presented that high-risk populations who perceive high pressure of significant others to get vaccinated against influenza are more likely to accept the vaccination [44,56,57]. Moreover, Chu et al. [20] identified that family was the primary influencer in individuals’ decision to receive the influenza vaccine during COVID-19. During the pandemic, people were motivated to take the influenza vaccine in order to protect their family members and friends [54]. Moreover, expectation from family or friends to accept the influenza vaccine during the pandemic was an important predictor of vaccine acceptance [17]. It seems to be that people who acknowledge the social benefit of the vaccination are more likely to get vaccinated against influenza.

We found that participants were more likely to accept the flu vaccine if they had received a COVID-19 booster dose in the past. Booster vaccination among participants indicates an even more positive attitude toward vaccination. Several studies confirm the positive relationship between a positive attitude toward vaccination in general and willingness to accept the flu vaccine during the COVID-19 pandemic [13,17,18,19,47]. Moreover, a systematic review found that historical vaccine acceptance was an important determinant of vaccination intention against influenza for the 2020–2021 season [21]. In particular, intention to receive a COVID-19 vaccine and previous influenza vaccination behavior are strong predictors of influenza vaccine willingness. Moreover, a systematic review found that vaccination uptake against H1N1 pandemic influenza was higher among those having been vaccinated in the past against seasonal influenza [26]. In addition, booster vaccination in our study could be an indicator of higher levels of fear against COVID-19 among high-risk populations. Literature suggests that fear of COVID-19 has a positive effect on vaccination intentions of influenza and COVID-19 [54,58]. It is probable that the burden of COVID-19 increases perceived risk and perceived susceptibility to influenza among high-risk populations. Moreover, our participants may consider that a possible co-infection with both COVID-19 and influenza could have poor outcomes for them. Moreover, two systematic reviews confirm the fact that perceptions of risk is a strong predictor of vaccination uptake against a variety of diseases including influenza [53,59]. 

Our study found a positive relationship between higher age and vaccination willingness. A systematic review including studies with individuals with chronic medical conditions confirm that older age is a promoter for influenza vaccine uptake [53]. Moreover, older age was related with higher intentions and uptake of the vaccine against H1N1 influenza [26]. It is reasonable that older people experience more fear of influenza since it is well-known that people over 65 years with comorbidities are at increased risk of severe influenza-related illness, hospitalization, and death due to influenza [1,60].

### Limitations

Our study had several limitations. First, we used a convenience sample that could not be representative of high-risk populations in Greece. For example, in our study most participants were females and possessed a university degree. Although we achieved the minimum required sample size, the extension of our results should be made with caution. Further studies with nationally representative samples could limit this selection bias. Second, we measured self-reported vaccination willingness instead of uptake. Therefore, actual vaccination behavior could be different from willingness. Future studies estimating the influenza vaccine uptake could give the opportunity to compare willingness and uptake rate. Third, we performed a cross-sectional study and our findings reflect influenza vaccination willingness at the specific time point of data collection. Future longitudinal studies could add information regarding individuals’ attitudes over time. Fourth, we investigated the impact of a variety of predictors (i.e., demographic characteristics, COVID-19-related variables, resilience, social support, anxiety, depression, and COVID-19-related burnout) on vaccination willingness but several additional predictors could be explored, e.g., kind of disease, media influence, access barriers, vaccine hesitancy, etc. Fifth, we assessed people’s intention to take the seasonal influenza vaccine. Perhaps, such intention could be different toward new vaccines against influenza or a possible pan-respiratory viruses vaccine (e.g., one vaccine annually protecting from SARS-CoV-2, influenza, and respiratory syncytial virus). Further research should investigate attitudes of high-risk groups toward new vaccines. Finally, selection bias is possible since people with limited Internet access cannot participate in our study. Moreover, people who are less vaccine hesitant and more research-minded are more likely to complete our study questionnaire. Thus, the willingness rate in our study could be an overestimation of the true rate.

## 5. Conclusions

We found that the intention of high-risk populations to receive the influenza vaccine was low. Moreover, our study contributes to an increased understanding of the factors that affect willingness of high-risk populations to accept the influenza vaccine. Policy makers could use this information to update vaccination programs against influenza. Identification of factors that affect vaccination intention is necessary to develop interventions to increase influenza vaccination uptake. Additionally, since COVID-19-related burnout and side effects from COVID-19 vaccination decrease the intention of people to accept the influenza vaccine, a jointly administered COVID-19 and influenza vaccination program could be developed and implemented. Emphasis should be given on safety and effectiveness issues since vaccine hesitancy increased during the pandemic.

Identification of factors that influence decision of high-risk populations to accept the influenza vaccine is crucial especially after relaxation of public health and social measures during the COVID-19 pandemic. Thus, influenza circulation in the winter 2022–2023 is expected to be high causing a longer and sharper epidemic. In particular, a modeling study using surveillance data from 11 countries in 2017–2022 found a 10–60% increase in the population susceptibility which might lead to a maximum of one-fold to four-fold rise in the epidemic size for the flu season 2022–2023 [61]. Moreover, early data from the 2022 influenza season in Australia showed that a larger than usual influenza outbreak should be expected in Europe [62]. These findings highlight the importance of achieving a high influenza vaccination coverage rate across countries. Our results could help policy-makers to develop proactive one-off influenza vaccination programs in order to reduce influenza circulation and influenza virus infections in the community. Public health authorities should apply measures tailored specifically for high-risk populations in order to decrease vaccine hesitancy. For example, specific educational programs should offer accurate knowledge about vaccines, increase trust in government and healthcare professionals, and eliminate conspiracy beliefs. Moreover, since high-risk populations are at higher risk of developing severe influenza disease, policy-makers should highlight the perceived benefits of vaccination against influenza and COVID-19.

High-risk populations should make a conscious decision to take the influenza vaccine annually. Transparent communication, use of facilitators to support influenza vaccination, and public awareness of the benefits of influenza vaccination can help increase vaccination coverage among high-risk populations. Continued and pertinent health education can change people’s attitudes and behavior and increase adoption of preventive measures such as vaccination against influenza.

## Figures and Tables

**Table 1 vaccines-11-00331-t001:** Demographic and COVID-19-related variables of the study population (*n* = 254).

Variables	*n*	%
Gender		
Males	54	21.3
Females	200	78.7
Age (years), mean, standard deviation	44.9	13.6
Educational level		
High school	56	22.0
University degree	198	78.0
Self-perceived health status		
Very poor	0	0
Poor	4	1.6
Moderate	46	18.1
Good	152	59.8
Very good	52	20.5
SARS-CoV-2 infection		
No	90	35.4
Yes	164	64.6
COVID-19 booster doses		
No	26	10.2
Yes	228	89.8
Adverse effects because of COVID-19 vaccination, mean, standard deviation	2.9	2.6

**Table 2 vaccines-11-00331-t002:** Descriptive statistics for the scales in the present study.

Scale	Mean	Standard Deviation	Median	Minimum Value	Maximum Value
Brief Resilience Scale	3.41	0.74	3.33	1.83	5.00
Multidimensional Scale of Perceived Social Support					
Family support	5.84	1.41	6.25	1.00	7.00
Friends support	5.61	1.52	6.00	1.00	7.00
Significant others support	5.94	1.40	6.50	1.00	7.00
Patient Health Questionnaire-4					
Anxiety	2.41	1.88	2.00	0.00	6.00
Depression	2.26	1.81	2.00	0.00	6.00
COVID-19 burnout scale					
Emotional exhaustion	3.41	1.21	3.60	1.00	5.00
Physical exhaustion	2.53	1.16	2.25	1.00	5.00
Exhaustion due to measures against the COVID-19	3.36	1.30	3.50	1.00	5.00
Influenza vaccination willingness	5.10	3.79	5.00	0.00	10.00

**Table 3 vaccines-11-00331-t003:** Univariate and multivariable linear regression analysis with influenza vaccination willingness score as the dependent variable.

Independent Variables	Univariate Model		Multivariable Model	
	Unadjusted Coefficient Beta (95% CI)	*p*-Value	Adjusted Coefficient Beta (95% CI) ^a^	*p*-Value
Gender (females vs. males)	1.204 (−2.333 to −0.075)	0.037	−0.364 (−1.488 to 0.760)	0.524
Age (years)	0.068 (0.035 to 0.101)	<0.001	0.040 (0.002 to 0.078)	**0.037**
Educational level (university degree vs. high school)	0.420 (−0.703 to 1.543)	0.462	0.377 (−0.736 to 1.491)	0.505
Self-perceived health status (good/very good vs. very poor/poor/moderate)	−0.419 (−1.572 to 0.735)	0.476	−0.115 (−1.279 to 1.049)	0.846
SARS-CoV-2 infection (yes vs. no)	0.138 (−0.836 to 1.112)	0.780	0.409 (−0.511 to 1.329)	0.382
COVID-19 booster doses (yes vs. no)	2.727 (1.228 to 4.227)	<0.001	1.841 (0.333 to 3.350)	**0.017**
Adverse effects because of COVID-19 vaccination	−0.361 (−0.535 to −0.187)	<0.001	−0.211 (−0.407 to −0.016)	**0.035**
Resilience	−0.003 (−0.634 to 0.628)	0.992	−0.311 (−1.063 to 0.441)	0.416
Family support	0.360 (0.034 to 0.686)	0.031	0.642 (0.181 to 1.103)	**0.007**
Friends support	−0.206 (−0.512 to 0.101)	0.187	−0.293 (−0.696 to 0.110)	0.154
Significant others support	−0.123 (−0.457 to 0.211)	0.467	−0.318 (−0.836 to 0.200)	0.228
Anxiety	−0.027 (−0.275 to 0.220)	0.828	0.353 (−0.031 to 0.737)	0.071
Depression	−0.070 (−0.329 to 0.188)	0.593	−0.209 (−0.628 to 0.210)	0.327
COVID-19-related emotional exhaustion	−0.195 (−0.579 to 0.189)	0.318	−0.038 (−0.597 to 0.520)	0.893
COVID-19-related physical exhaustion	0.053 (−0.349 to 0.455)	0.794	0.423 (−0.189 to 1.036)	0.174
Exhaustion due to measures against the COVID-19	−0.512 (−0.866 to −0.159)	0.005	−0.561 (−0.986 to −0.137)	**0.010**

Bold *p*-values indicate statistically significant associations in the multivariable model. CI: confidence interval. ^a^ *p*-value for ANOVA < 0.001; R^2^ for the final multivariable model was 14.5%.

## Data Availability

The data presented in this study are available on request from the corresponding author.

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
