# Peer review of "Predictors of Seasonal Influenza Vaccination Willingness among High-Risk Populations Three Years after the Onset of the COVID-19 Pandemic"

_vaccines, 2023, doi:10.3390/vaccines11020331_

Round 1

Reviewer 1 Report

The study is well done and the results are presented in an adequate way. have only three points:

1) page 3 l 147: Test for Normality is neither  necessary (high sample size) nor appropriate (in regression one should check the residuals). just leave out this sentence.

As the authors correctly stat, the variable "COVID-19 booster doses" is related to willingness for vaccination. This is also seen in the rather strong resulting effect. However, to have such a variable in the multivariable model could be problematic. Possible effects of other variables could be concealed. I would propose to do a sensitivity analysis without this variable.     

Author Response

Dear Reviewer, we are grateful for your comments. You really help us to improve our manuscript. We apply all your suggestions to our manuscript. Also, we made changes in the manuscript according to the other Reviewers’ instructions. We hope that our revision will reach the high standards of the journal “Vaccines”. 

We look forward to hearing from you

Best Regards

The authors

Reviewer 2 Report

This interesting paper by Katsiroumpa and colleagues aims to estimate the willingness of high-risk populations to take the influenza vaccine for the 2022-2023 season, and to identify factors that influence such willingness. 

This is an important topic as influenza vaccination coverage rates (IVCR) in Europe do not reach the target set by WHO and WHO Europe (with few exceptions). Therefore, understanding the factors that influence the willingness to be vaccinated against flu could be helpful to increase IVCRs. 

Some issues I recommend to consider while revising the paper: 

- the introduction is lacking important elements, and therefore I suggest: 

* specifying the study is conducted in Greece (line 63)

* considering a revision of the paragraph on the impact of influenza vaccination (IV) on COVID-19. The studies conducted so far (including SR and MA) are not free from important biases and any conclusions should be drawn with care, with a more conservative approach. I suggest specifying that there is no evidence that IV can increase the risk of SARS-CoV-2 infection or severe outcomes (as hypotesized in the past) and that it may even have a positive impact (to be confirmed by solid studies). 

* adding background information about IVCR in Europe and Greece should be added (some examples: 1-2) along with information about willigness to get vaccinated (that you correctly reported). 

* better explaining why it is so important to understand the factors that promote the willigness to get vaccinated (e.g. sometimes they are actionable items). 

- Methods are explained and results presented pretty well

- In the discussion, I suggest better exploring the public health impact of your findings, e.g. by comparing them with two important studies that tried (apparently successfully) to understand the impact of low influenza circulation and vaccine coverage rates on the current influenza season (3-4).

One last question: have you assessed the positive/negative response of your sample towards the possibility of new vaccines against influenza? 

E.g. whether they would be more willing to vaccinate with a more effective vaccine, or with a pan-respiratory viruses vaccine (e.g. one shot every year protecting from SARS-CoV-2, Influenza, RSV)

1. https://pubmed.ncbi.nlm.nih.gov/34915972/

2. https://pubmed.ncbi.nlm.nih.gov/35632553/

3. https://pubmed.ncbi.nlm.nih.gov/36240828/

4. https://pubmed.ncbi.nlm.nih.gov/36578202/

Author Response

(The authors gave the same response as above.)

Round 2

Reviewer 2 Report

Dear Authors, 

many thanks for addressing all my comments. 

Just one minor thing: 

- there was no misunderstanding about the effect of influenza vaccination on the risk of sars-cov-2 infection or severe covid-19 outcomes. I suggest just being more careful when saying that influenza vaccination could have a good/positive impact (reduce risks) as certainly - at the moment - the evidence is not sufficient to affirm that. I would stress this uncertainty by saying that it might be beneficial also in reducing these risks, but more (and bigger, and hopefully prospective ones) studies are needed to draw any conclusions. 

Best, 

Author Response

Dear Reviewer, thank you again for your time and your comment. Our apologies for our misunderstanding. We re-write the paragraph according to your comment. Now the text is the following:

"Literature suggests that influenza vaccination might be beneficial on SARS-CoV-2 infection and clinical outcomes among COVID-19 patients. In particular, meta-analyses showed that influenza vaccination reduced the risk of SARS-CoV-2 infection up to 24%, of hospitalization up to 29%, of intensive care unit admission up to 37%, and of death up to 32% [9–12]. However, we should notice that the evidence is not sufficient to affirm the positive effect of influenza vaccination. Prospective, bigger, and more valid studies should be conducted in order to get more solid results."